# Research on the speed thresholds of trucks in a sharp turn based on dynamic rollover risk levels

**Tian Xin**[1], **Jinliang Xu**[1]*, **Chao Gao**[1], **Zhenhua Sun**[2]

**1** School of Highway, Chang'an University, Xi'an, Shaanxi, China, **2** Shaoxing Communications Investment Group Co., Ltd., Shaoxing, Zhejiang, China

* xujinliang@chd.edu.cn

**Data Availability Statement:** All relevant data are within the manuscript.

**Funding:** This research was supported by Scientific Research Project of Zhejiang Provincial Department of Transportation (No. 2020025) to JX and ZS. This project was completed by the

## Abstract

Truck rollover is a problem that seriously endangers the safety of human life. Under special conditions, when the driver takes a sharp turn, the truck is most prone to rollover. Speed seriously affects the driving stability of the truck in a sharp turn, but the calculation of the safe speed is not accurate enough at present. The aim of this paper is to develop a more accurate safe speed calculation method to avoid the truck rollover in a sharp turn. Firstly, the calculation formula of the rollover threshold was derived based on a theoretical model, then, the simulation tests were carried out. We selected a 4-axle truck with a total weight of 30t as the subject, simulated the dynamic process of the truck rollover in a sharp turn with TruckSim, evaluated the dynamic rollover risk levels of the truck during this process, and verified the accuracy of the simulation results by results of the theoretical model. Finally, by analyzing the steering principle of the vehicle, the safe speed threshold and the limit speed threshold of the truck in a sharp turn were calculated according to the lateral acceleration corresponding to the rollover risk levels. The results show that no matter what the loading condition of the truck is, when the rollover margin is reduced to about 0.15g, the truck just reaches the risk level of critical rollover; the result provides an accurate algorithm for speed thresholds of the truck when turning radius is less than 250 m. The research provides a calculation method for safe speed of trucks from a dynamic perspective. The research results can be applied to the speed warning system of trucks, which can make drivers better control the rollover risk of trucks in the process of driving and improve driving safety.

## 1. Introduction

Truck rollover is one of the most serious accident forms that cause traffic casualties.

The road interruption caused by truck rollover and the environmental pollution caused by the leakage of chemicals and dangerous goods in the process of truck rollover also cannot be ignored [1–3]. From the 1950s to the present, researchers have established a full range of prevention and control system, including road safety facilities, vehicle safety systems, risk prevention and control, and safety education in the research areas such as vehicle structure, road

cooperation between Chang'an University and Shaoxing Communications Investment Group Co., Ltd. The funder provided support in the form of salaries for authors, but did not have any additional role in study design, data collection and analysis, decision to publish, or preparation of the manuscript. The specific role of these author is articulated in the 'author contributions' section.

**Competing interests:** The authors have declared that no competing interests exist. There are no patents, products in development or marketed products association with this research to declare. This does not alter our adherence to PLOS ONE policies on sharing data and materials.

engineering, traffic engineering, laws and regulations, etc. [4–9]. Although we have made great efforts in the study of traffic safety, a frequently reported vehicle rollover accidents show that we do not have a thorough understanding of the mechanism of vehicle rollover accidents.

Due to the inherent complex nature of road traffic safety, based on current understanding, it is generally believed that the interaction of driver-vehicle-road-environment is the ultimate cause of vehicle rollover [10, 11]. The driver is the most important control factor in the road traffic system. Studies and practical evidence from countries around the world show that more than 90% of traffic accidents are related to unreasonable or wrong driving behaviors to varying degrees [12–15]. Among the unreasonable or wrong driving behaviors, the more common ones are speeding, sharp turning, and sudden braking. The study used the three years accident data in North Carolina to analyze the impact of driver behavior, vehicle and road factors on the severity of truck rollover and injury to occupants. The results showed sharp turning and turning movements were associated with a higher rollover risk [16, 17]. The Sharp turning under special conditions will make the actual trajectory of vehicle inconsistent with the road design trajectory, which will seriously affect the driving stability. In road design, it is assumed that the lateral force coefficient "$\mu$" is a function of the three factors of speed, circular curve radius, and superelevation as variables, and that the vehicle is traveling on the circular curve strictly following the established trajectory, and it is assumed that the vehicle makes uniform circular motion [18]. This design theory was adopted in road design by countries all over the world. In the mid-1990s, scholars evaluated the safety of this design criterion. They believe that when using existing standards to guide the geometric design of roads, as long as the selected design speed is in line with reality circumstances, then roads designed according to existing theories can ensure that passenger cars and trucks do not have rollover or sideslip accidents. However, they further pointed out that for trucks, if their tires are relatively worn and their rollover thresholds are small, or the road is wet and damaged, even if the minimum radius of the curve is used as the design value under the condition of a design speed of only 10 or 20 mph, the safety of driving may not be guaranteed [19]. Scholars further emphasized as long as the driving speed selected by the driver is less than or equal to the design speed, the road design indicators guided by the existing design theory can ensure neither trucks nor passenger cars will skid or rollover, so as to ensure driving safety [20, 21]. However, this conclusion is based on the following idealized conditions: 1) The vehicle is simplified to a point mass model. 2) When a vehicle passes through a circular curve section, the radius of curvature of each point on its trajectory is equal to the designed circular curve radius.3) The vehicle makes uniform circular motion. But in fact, the driver will inevitably encounter various emergencies during the driving process, such as throwing objects in front of the vehicle, parking on the side of the road, obstructed vision, etc., and they need to take emergency maneuvers such as sharp turning and lane change etc.

Speeding will increase the risk of vehicle rollover in a sharp turning, especially for trucks. There are many studies on the safe speed on the curve section. The UK External Vehicle Speed Control (EVSC) project emphasized the best prediction of accident reduction was that the fitting on all vehicles of a simple mandatory system, with which it would be impossible for vehicles to exceed the speed limit, would save 20% of injury accidents and 37% of fatal accidents [22]. The development of Intelligent Transportation Systems (ITS) provides some new ideas for solving the curve negotiation issue. One solution to improve the warning accuracy is the vehicle-mounted Curve Speed Warning (CSW) system. Various curve speed models are used in CSW to judge whether a vehicle approaches curves too fast [23]. The research proposed an elaborate model that fully considered the road geometry and vehicle parameters for the calculation of the maximum safe speed when approaching an upcoming curve [24–26]. But so far, vehicle anti-rollover speed control system on curves, the algorithm core of its safe speed is

mainly to simplify the vehicle into a rigid body to analyze, with rarely involving the characteristics of suspension and tires [27–29].

Some studies analyzed that it increases the difficulty of vehicle control when traveling on a small radius curve, because excessive lateral acceleration (related to the lateral force coefficient) causes the driver to feel uncomfortable when turning. Once the lateral acceleration reaches a critical level, the vehicle will face risk of rollover [30]. Some research results indicated that when the radius of the circular curve is less than 350 m and the height of center of gravity is greater than 2 m, the truck with a small track width is more likely to rollover [31]. Studies have found that superelevation has a significant impact on the lateral acceleration of the vehicle, and the lateral acceleration is a sensitive parameter that causes the vehicle to rollover [32]. Studies have pointed out that when evaluating the risk of vehicle rollover based on lateral acceleration indicators, horizontal curve radii, superelevation and their interaction have a highly significant impact on the vehicle rollover [33].

The above research results show that the radius of curvature, the superelevation, the height of the center of gravity and driver behaviors have a significant impact on vehicle rollover. However, in the majority of previous studies, the radius of curvature referred to the horizontal curve radius of the road geometry itself, rather than the radius of curvature of actual trajectory curve of the vehicle. Under special conditions, the radius of curvature of actual trajectory curve of the vehicle may be much smaller than the radius of curvature of road horizontal curve. Secondly, because it is impossible to accurately reproduce the whole process of vehicle rollover, the calculation of safe speed is mainly based on the traditional static rollover model, which simplifies the vehicle as a rigid body, so that results are not reasonable and accurate.

In order to effectively reduce the hazards of truck rollover accidents on the road, the present study simulated the whole dynamic process of truck rollover with TruckSim, evaluated the dynamic rollover risk levels of the truck, and verified the accuracy of the simulation results by calculation results of the theoretical model. Finally, the safe speed threshold and the limit speed threshold was calculated. The research provides a method for calculating the safe speed of a truck in a sharp turn from a dynamic perspective, which can be used for the truck speed warning system.

The remaining parts of this article are organized as follows: Section 2 introduces a theoretical model of the rollover threshold, and then carry out simulation tests and obtains data, evaluates the dynamic rollover risk levels, verifies the accuracy of the simulation results by calculation results of the theoretical model and calculate the safe speed threshold and the limit speed threshold. Section 3 is the results and discussion. Section 4 summarizes the main conclusions of this research.

## 2. Methods

### 2.1. A rollover threshold formula based on the two-track model

**2.1.1. Basic assumptions of the two-track model with a simple suspension structure.**   In order to facilitate the analysis, we make the following assumptions about the vehicle model:

1. the input of the left and right wheels are the same, the left and right sides of the vehicle move in exactly the same way;

2. the wheel is a rigid body;

3. the height of the center of gravity is the normal height of the center of gravity, different from the assumption of the point mass model that the height of the center of gravity is on the road surface;

4. the vehicle has a non-rolling unsprung mass composed of axles and wheels, and a sprung mass with degree of freedom of roll.

**2.1.2. Rollover threshold equation.**  We first introduce the definitions of two key terms in this paper.

**Rollover threshold** $\mu_{y,rollover}$: the maximum lateral acceleration that a vehicle can experience without overturning.

**Rollover Margin** $RM_{ay}$: it is based on lateral acceleration, which represents the difference between the current lateral acceleration and the maximum lateral acceleration that a vehicle can experience without overturning.

Then we establish a rollover model considering a simple suspension. This model mainly refers to the research of Torbic et al. [34].

It can be assumed that $F_{zi} = 0$ just before wheel lift occurs, it is as shown in Fig 1. At this time, a moment balance equation is established for the contact point of the outer wheel:

$$\Sigma M = -h_g \left( m \frac{v^2}{R} - mg \, sin \, \alpha \right) + \left( \frac{T}{2} - (h_g - h_r)\emptyset \right) mg \, cos \, \alpha = 0 \tag{1}$$

Where: $T$-Track width(m) $h_g$- Height of center of gravity(m). $h_r$-Height of the rollover center, (m). $F_{zi}$-Normal force of inner tire(m). $\varphi$-Roll angle(m). $\alpha$-Superelevation slope(˚). $v$-Speed(m/s). $R$-Radius of circular curve, m. $g$-Gravitational acceleration (9.81m/s$^2$). $m$-Sprung mass(kg).

where $a_y = v^2/R$, substituting it into Eq (1), and using the small angle approximation yields, Eq (1) can be rewritten as the following form:

$$-h_g \left( \frac{a_y}{g} - i_h \right) + \left( \frac{T}{2} - (h_g - h_r)\varphi \right) = 0 \tag{2}$$

Where: $i_h$-Superelevation rate. Eq (2) is slightly deformed, then Eq (2) becomes:

$$\frac{a_y}{g} + \frac{(h_g - h_r)}{h_g} \varphi = \frac{T}{2h_g} + i_h \tag{3}$$

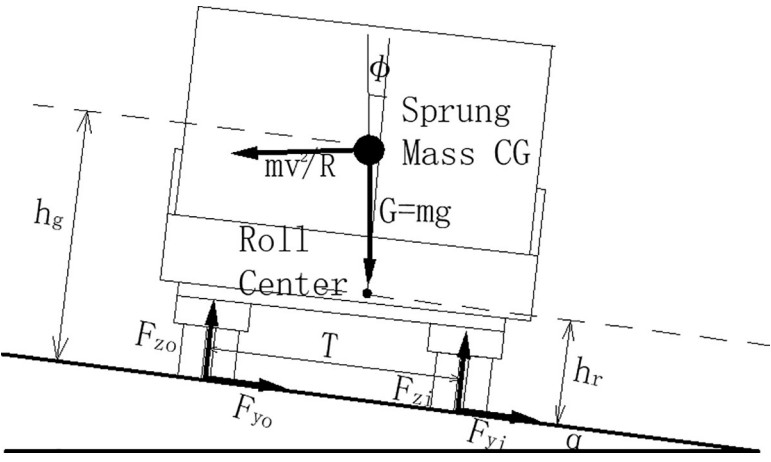

**Fig 1. Vehicle rollover model including superelevation.**

The roll angle of the vehicle body can be written as a roll gain, in rad/g, multiplied by the lateral acceleration, introducing a coefficient $R_\varphi$, $\varphi = R_\varphi a_y/g$, Eq (3) can be rewritten as:

$$\frac{a_y}{g} = \frac{\frac{T}{2h_g} + i_h}{1 + \left(1 - \frac{h_r}{h_g}\right)R_\varphi} \tag{4}$$

From Eq (4), it can be easily found that rollover depends only on the magnitude of the lateral force. The Longitudinal slope and the speed will not affect the rollover threshold. For vehicles with suspension systems, the most unfavorable situation is that the vehicle has a large roll angle and the very low roll axis, that is, $\frac{h_r}{h_g} = 0$ or negative, it is the same as our intuitive feeling. For example, a heavy vehicle with a high load seems to be equipped with a soft suspension system.

In addition, if the vehicle is a rigid body without a suspension system, $R_\varphi = 0$, then Eq (4) becomes:

$$\frac{a_y}{g} = \left(\frac{T}{2h_g} + i_h\right) \tag{5}$$

For this rigid vehicle model, increasing the superelevation rate can directly increase the rollover threshold. It is also consistent with the current design code that small radius section should increase the superelevation rate. For vehicles without suspension or roads without superelevation, the rollover threshold is $T/2h_g$, this is the common expression of the vehicle static rollover threshold.

In order to approximate the rollover threshold of a simple suspension vehicle model, according to the previous test data, when the rollover acceleration is 0.1g, the roll angle of the vehicle is 1°, that is $R_\varphi = 10°/g$或0.17rad/g. For most vehicles, there is $h_r/h_g = 0.25\sim0.75$ [35, 36]. The worst case is when this value is 0, and at the same time, there is no superelevation, use $RM_{ay}$ to represent the rollover margin, it can be deduced:

$$RM_{ay} = \frac{\frac{T}{2h_g}}{1 + R_\varphi} - \frac{a_y}{g} \approx \frac{1}{1.17}\left(\frac{T}{2h_g}\right) - \frac{a_y}{g} \approx 0.85\left(\frac{T}{2h_g}\right) - \frac{a_y}{g} \tag{6}$$

And therefor, $\mu_{y,rollover}$ can be rewritten in the final form used in this study:

$$\mu_{y,rollover} = \frac{a_y}{g} = 0.85\left(\frac{T}{2h_g}\right) \tag{7}$$

Where: $a_y/g$-Normalized acceleration within curve.

## 2.2. Simulation tests to evaluate the dynamic risk levels of truck rollover

**2.2.1. Simulation platform and simulation purpose.** In this study, TruckSim2016.1 was used to simulate the dynamics and kinematics of trucks. TruckSim is a dynamic simulation software developed by the Mechanical Simulation Corporation (MSC), based on years of testing and vehicle dynamics research experience of UMTRI, University of Michigan Highway Transportation Research Institute. It is used for simulate and analyze the dynamic characteristics of light trucks, buses, heavy trucks, multi-axle semi-trailers, etc. It has been applied to the development and research of trucks, military vehicles and other special-purpose vehicle systems by many companies and universities, including United States General Motors, Sweden Volvo, Germany Benz and Japan Mitsubishi, South Korea Hyundai, University of Michigan,

Jilin University and Zhejiang University. The purpose of the simulation was to obtain the rollover threshold of the truck and the maximum lateral acceleration when the truck is at a risk level of critical rollover.

**2.2.2. Evaluation indicators and evaluation standards of rollover risk.** The vehicle rollover indicator is an indicator quantity that describes the dangerous degree of vehicle rollover at any time. For trucks with high speeds and heavy loads, almost all trucks will rollover when tires on one side of the truck rise because of the existence of greater motion inertia. Exceptions will only happen under special conditions, as experienced racers control the specially developed vehicle, the vehicle will return to a steady state when tires on one side of the vehicle rise. Therefore, this study selected the vertical reaction forces of the tires on one side of the truck as the evaluation indicator, the evaluation standards of rollover risk levels were as follows:

1. the vertical reaction force of neither of tires was equal to zero, which meant that wheel lift didn't occur and the truck was in a normal driving state. Then the rollover risk level was level 1, which was defined as "safety";

2. the vertical reaction forces of any one or some of tires were equal to zero, it meant that there was one or some wheels rising, but not all wheels on one side of the truck rose. There was still the vertical reaction force between the tires and the ground. At this moment, the truck was in an extreme driving state, but it was still possible for the truck to return to the normal driving state. Then the rollover risk level was Level 2, which was defined as "critical rollover";

3. the vertical reaction forces of the tires on one side of the truck were all equal to zero, which meant that the wheels on one side of the truck all rose. At this moment, the truck might rollover at any time. Then the rollover risk level was level 3, which was defined as "rollover".

**2.2.3. Simulation model establishment.** *1) Vehicle modeling.* In this study, the models of typical 4-axle truck with three loading conditions were established, which were none-loaded (the body mass of none-loaded truck was 11t and the height of the center of gravity was 1.273m), partially-loaded (the load mass was 9.5t and the height of the center of gravity of the load was 1.75m) and fully-loaded (the load mass was 19t and the height of the center of gravity of the load was 2m).The prototype trucks was a FAW Jiefang J6M8X4 series trucks, which were chosen because they accounted for a large proportion of existing trucks in China, because of their operational characteristics, especially because they were prone to rollover. In order to "define" a truck, some input parameters of the truck were required. Through the combination of literature review and the default values found in the simulation software, the truck parameters were determined [37, 38]. In this way, the modeling of seven parts were completed including the vehicle shape, braking system, power transmission system, steering system, tires, suspension system and load. Due to limited space, we don't introduce each of them in detail. The appearances of trucks are shown in Fig 2.

*2) Road modeling.* In this study, the road geometry alignment was selected as a good one. The radius of the circular curve was selected as the minimum radius of 5500m without superelevation (V = 120km/h), the superelevation rate was set to zero, and the longitudinal gradient was set to zero [39]. Pavement was dry asphalt concrete, so the friction coefficient of the road was set to 0.70. The number of lanes was set as a two-lane, the width of each lane was set to 3.75 m with hard shoulders on both sides, and roadside was set as a lawn. Road simulation image is shown in Fig 3.

*3) Driving strategy modeling.* This study set a variety of target speeds and lane change trajectory curves with TruckSim. The truck traveled along a trajectory curve at a target speed. The

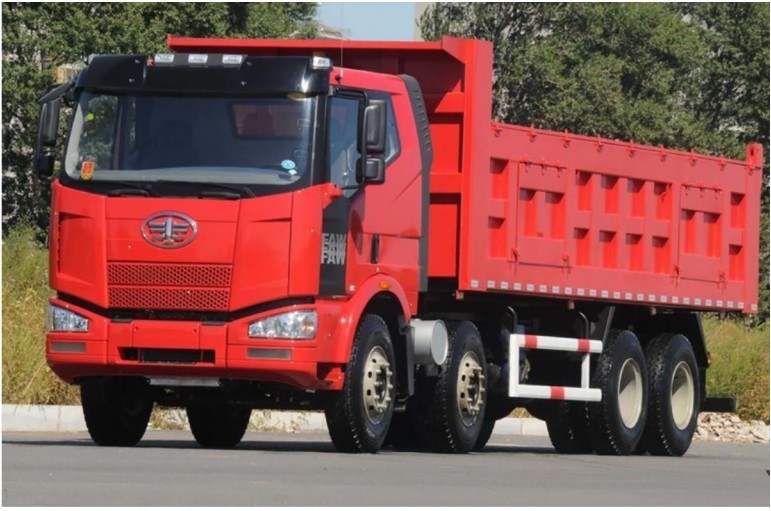

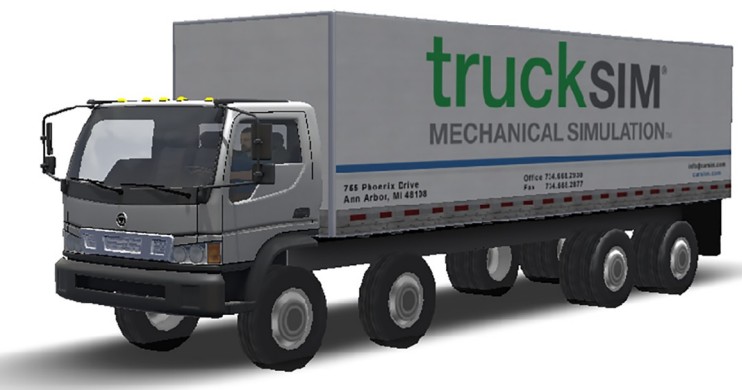

**Fig 2. Appearances of the trucks.** a) the prototype truck b) the simulation truck.

given trajectory was equivalent to the given steering information. When the driver received the steering information, he turned the steering wheel to perform steering operations.

Because the maximum speed of the prototype truck was 105 km/h, the target speeds were set to 105 km/h, 100 km/h, 90 km/h, 80 km/h, 70 km/h, and 60 km/h.

The preset lane change trajectory curves, which were numbered No.1, No.2, No.3, No.4, and No.5, required the driver to complete the operation of changing from the current lane to the adjacent lane, as shown in Fig 4. The truck started traveling along the centerline of the lane, the ordinate is -1.875m, and gradually shifted laterally to the position where the ordinate is 1.875m, that is, shifted to the position of the centerline of the adjacent lane, the total offset is one lane width 3.75m. This completed the lane change.

The abscissa of Fig 4 represents the road station information, that is, the distance the truck travels in the longitudinal direction. The No.1 curve is a general lane change trajectory curve, and curves from No.2 to No.5 are emergency lane change trajectory curves. From Fig 4, we can clearly find out that as the number go from 1 to 5, lane change behavior is getting more and more rapider, and the time for the lane change process is getting more and more shorter.

**2.2.4. Simulation scenes construction and demonstration.** A total of 39 scenes were simulated in this study, the scene construction mainly involved load elements, road elements, and

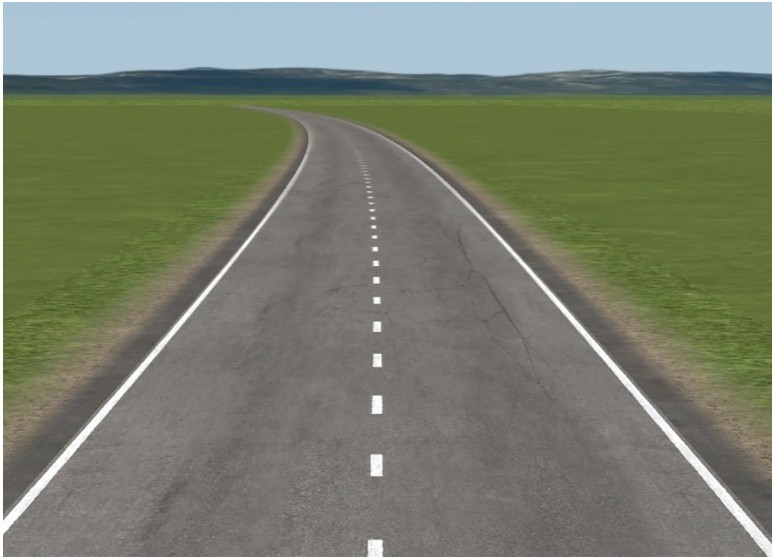

**Fig 3. Road simulation image.**

driver behaviors elements. The specific parameter settings of each element have been completed in vehicle modeling, road modeling, and driving strategy modeling. The scene construction was an orthogonal combination of various elements. The specific characterization of parameters of simulation scene is shown in Table 1.

The scenes construction process was as follows in general. First, trucks under three loading conditions respectively followed the No. 1 lane change trajectory curve at the speed of 105km/h. It was found that the wheel lift didn't occur under three loading conditions.

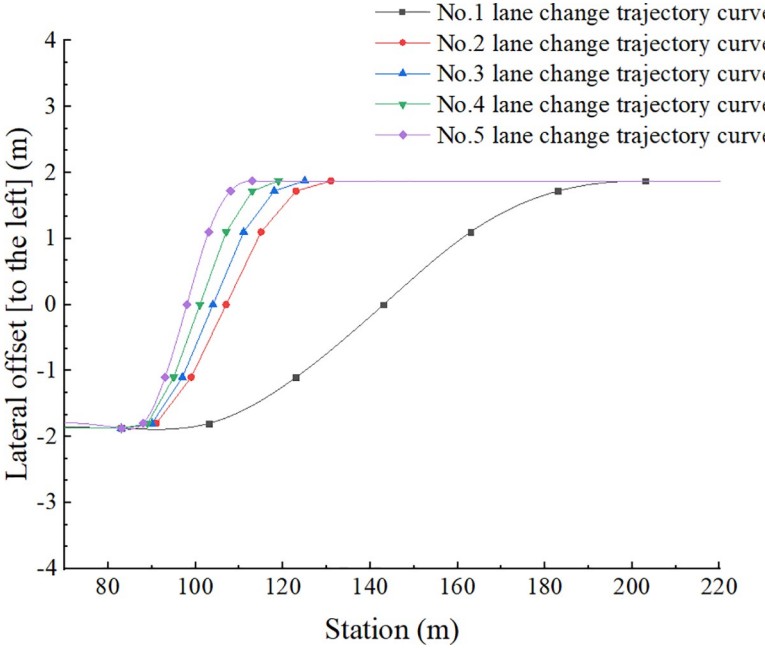

**Fig 4. Parameters of the given trajectory curves.**

**Table 1. Characterization of parameters of simulation scenes.**

| Scenes Construction | Road features | | | | Load parameters | | Driver behaviors | |
|---|---|---|---|---|---|---|---|---|
| | Radius of Circular Curve | Superelevation rate | Longitudinal gradient | Friction Coefficient | Load Mass | Height of the center of gravity | Speed | Lane change trajectory curve |
| | (m) | (%) | (%) | | (kg) | (m) | (km/h) | Number |
| Scene1 | 5500 | 0 | 0 | 0.7 | 0 | 0 | 105 | 1 |
| Scene 2 | 5500 | 0 | 0 | 0.7 | 9500 | 1.75 | 105 | 1 |
| Scene 3 | 5500 | 0 | 0 | 0.7 | 19000 | 2 | 105 | 1 |
| Scene 4 | 5500 | 0 | 0 | 0.7 | 0 | 0 | 105 | 2 |
| Scene 5 | 5500 | 0 | 0 | 0.7 | 9500 | 1.75 | 105 | 2 |
| Scene 6 | 5500 | 0 | 0 | 0.7 | 19000 | 2 | 105 | 2 |
| Scene 7 | 5500 | 0 | 0 | 0.7 | 0 | 0 | 100 | 2 |
| Scene 8 | 5500 | 0 | 0 | 0.7 | 9500 | 1.75 | 100 | 2 |
| Scene 9 | 5500 | 0 | 0 | 0.7 | 19000 | 2 | 100 | 2 |
| Scene 10 | 5500 | 0 | 0 | 0.7 | 0 | 0 | 100 | 3 |
| Scene 11 | 5500 | 0 | 0 | 0.7 | 9500 | 1.75 | 100 | 3 |
| Scene 12 | 5500 | 0 | 0 | 0.7 | 19000 | 2 | 100 | 3 |
| Scene 13 | 5500 | 0 | 0 | 0.7 | 0 | 0 | 100 | 4 |
| Scene 14 | 5500 | 0 | 0 | 0.7 | 9500 | 1.75 | 100 | 4 |
| Scene 15 | 5500 | 0 | 0 | 0.7 | 19000 | 2 | 100 | 4 |
| Scene 16 | 5500 | 0 | 0 | 0.7 | 0 | 0 | 100 | 5 |
| Scene 17 | 5500 | 0 | 0 | 0.7 | 9500 | 1.75 | 100 | 5 |
| Scene 18 | 5500 | 0 | 0 | 0.7 | 19000 | 2 | 100 | 5 |
| Scene 19 | 5500 | 0 | 0 | 0.7 | 0 | 0 | 90 | 2 |
| Scene 20 | 5500 | 0 | 0 | 0.7 | 9500 | 1.75 | 90 | 2 |
| Scene 21 | 5500 | 0 | 0 | 0.7 | 19000 | 2 | 90 | 2 |
| Scene 22 | 5500 | 0 | 0 | 0.7 | 0 | 0 | 80 | 2 |
| Scene 23 | 5500 | 0 | 0 | 0.7 | 9500 | 1.75 | 80 | 2 |
| Scene 24 | 5500 | 0 | 0 | 0.7 | 19000 | 2 | 80 | 2 |
| Scene 25 | 5500 | 0 | 0 | 0.7 | 0 | 0 | 70 | 2 |
| Scene 26 | 5500 | 0 | 0 | 0.7 | 9500 | 1.75 | 70 | 2 |
| Scene 27 | 5500 | 0 | 0 | 0.7 | 19000 | 2 | 70 | 2 |
| Scene 28 | 5500 | 0 | 0 | 0.7 | 0 | 0 | 60 | 2 |
| Scene 29 | 5500 | 0 | 0 | 0.7 | 9500 | 1.75 | 60 | 2 |
| Scene 30 | 5500 | 0 | 0 | 0.7 | 19000 | 2 | 60 | 2 |
| Scene 31 | 3000 | 2 | 0 | 0.7 | 0 | 0 | 105 | 3 |
| Scene 32 | 3000 | 2 | 0 | 0.7 | 9500 | 1.75 | 105 | 4 |
| Scene33 | 3000 | 2 | 0 | 0.7 | 19000 | 2 | 105 | 5 |
| Scene 34 | 2000 | 4 | 0 | 0.7 | 0 | 0 | 105 | 3 |
| Scene 35 | 2000 | 4 | 0 | 0.7 | 9500 | 1.75 | 105 | 4 |
| Scene 36 | 2000 | 4 | 0 | 0.7 | 19000 | 2 | 105 | 5 |
| Scene 37 | 1000 | 6 | 0 | 0.7 | 0 | 0 | 105 | 3 |
| Scene 38 | 1000 | 6 | 0 | 0.7 | 9500 | 1.75 | 105 | 4 |
| Scene 39 | 1000 | 6 | 0 | 0.7 | 19000 | 2 | 105 | 5 |

Therefore, the driving strategy was changed to follow the No. 2 lane change trajectory curve at a speed of 105km/h, and trucks under three loading conditions were tested according to this driving strategy, the wheel lift occurred. At this time, the trajectory curve was unchanged, the speeds were set to 100km/h, 90km/h, 80km/h, 70km/h, 60km/h

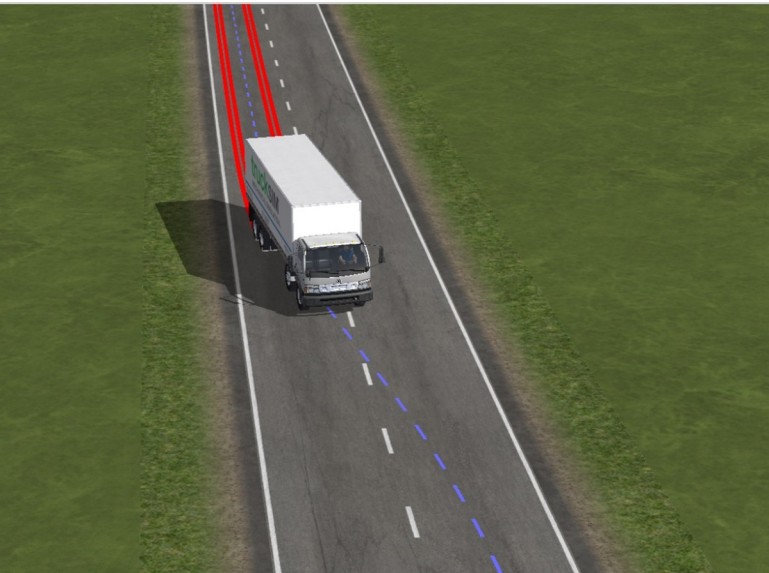

**Fig 5. Animation demonstration screenshot of scene 10.**

respectively and simulation tests on trucks under three loading conditions were conducted. Only at a speed of 60km/h, wheel lift didn't occur, and wheel lift occurred in other speed conditions. In order to obtain the rollover threshold of trucks under each loading condition, the speed was always set to 100km/h, and then the trajectory curve was continuously shifted, and it was found that when the trajectory cure is set to No. 3, tires on one side of the truck under full-loaded condition all rose. When the trajectory curve was set to No. 4, the tire on the side of the truck under partially-loaded all rose. When the trajectory curve was set to No. 5, the tire on the side of the truck under none-loaded condition rose. scenes 31–39 are used to test the rollover thresholds of the truck at a speed of 105km/h under different superelevation rates.

In this way, we had completed the simulation tests.

The simulation scene demonstration is shown in Fig 5 with taking scene 10 as an example. The blue dashed line is lane change trajectory curve, and the red marks are the trajectory of the tires on the axis 4.

**2.2.5. Test data processing.** The purpose of the simulation tests is to obtain the rollover threshold of the truck and the maximum lateral acceleration when the rollover risk level is level 2: "critical rollover".

According to the evaluation indicators and evaluation standards of rollover risk, we need to obtain the maximum lateral acceleration when the vertical forces of the tires on one side of the truck are all equal to zero; we need to obtain the maximum lateral acceleration when the vertical force of any tire is equal to zero.

The data visualization of TruckSim provides users with convenient and quick data reading channels. The time step of the simulation output results is 0.025s, which is enough to get the data change in a small time. The output results can be a graph or an excel table. The software can also directly give the maximum and minimum values of the results. In addition, the accuracy of the data is high. For example, the lateral acceleration is accurate to 0.000001g, and the wheel force is accurate to 0.1N. All of the above ensure the accuracy of the data obtained.

### 2.3. Predicting the speed thresholds of the truck in a sharp turn based on the dynamic rollover risk levels

**2.3.1. Steering principle of the vehicle.**    Three states of the vehicle classified according to turning behavior are shown in Fig 6A, 6B and 6C. When the vehicle enters a curved section from a straight section, the driver needs to turn the steering wheel. The front wheels are pressed to a certain degree through the rotation of the steering mechanism. It ensures that the vertical lines of the inner and outer front wheels and the vertical lines of the rear wheels intersect at the same point O. At this time, the vehicle will move in a circle around point O, which is called the turning center. The distance from point O to the center of the rear axle is the turning radius of the vehicle. When the vehicle reaches a steady state on a circular curve, the turning radius of the vehicle is equal to the radius R of the circular curve of the road, as shown in Fig 6B. Under special conditions, the driver takes a sharp turn, the front wheel angle of the vehicle suddenly increases from $\theta$ to $\theta'$, the turning center of the vehicle becomes O', and the turning radius of the vehicle suddenly decreases from R to R', as shown in Fig 6C. Under the following special conditions, such as a curve with a large radius immediately followed by a curve with a small radius, or throwing objects in front of the vehicle, parking on the side of the road, obstructed vision, etc., the driver will probably take a sharp turn. At these times the turning radii of the vehicle are often very small.

**2.3.2. Safe speed threshold and limit speed threshold.**    Excessive lateral force coefficient is the root cause of the vehicle rollover. When the driver takes a sharp turn, the turning radius

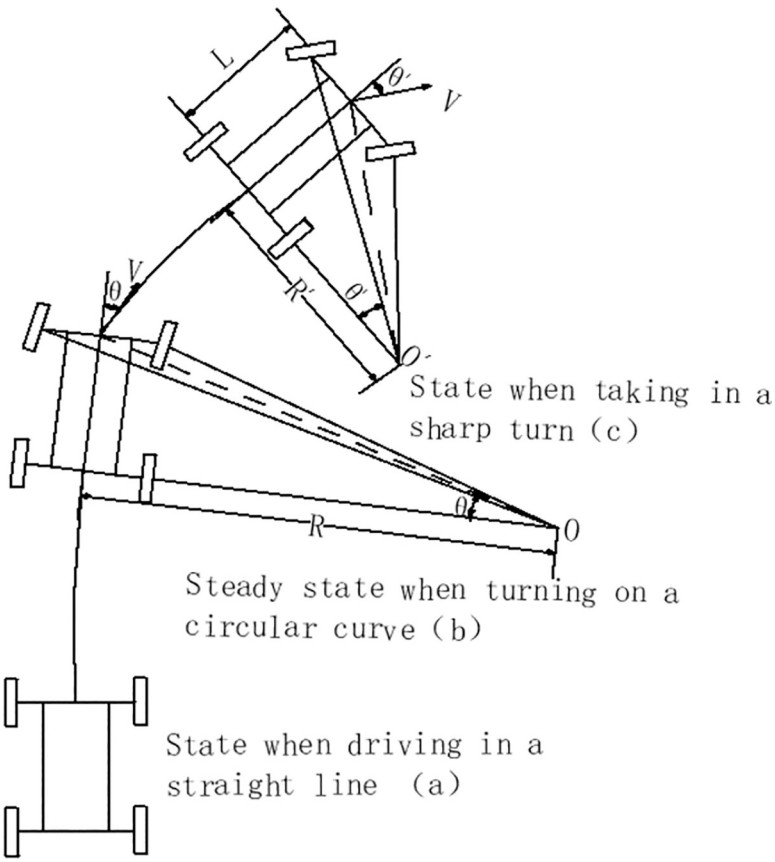

**Fig 6. Schematic diagram of vehicle steering principle.**

is suddenly reduced, and the lateral force coefficient increases sharply. According to the geometric relationship in Fig 3(C), the lateral force coefficient of the vehicle can be expressed as:

$$\mu_y = \frac{v^2}{gR'} - i_h = \frac{a_y}{g} - i_h = \frac{V^2}{127R'} - i_h \tag{8}$$

The superelevation can offset part of the lateral force coefficient, and the other part can be resisted by the roll stability of the vehicle, which depends on the vehicle design, load and suspension characteristics. In the previous section, with the help of the powerful function of TruckSim, we fully considered the comprehensive effects of vehicle design, load and suspension characteristics, conducted a dynamic analysis of the rollover process of the truck, and evaluated the dynamic rollover risk levels of the truck. Based on the dynamic rollover risk levels of the truck, the speed thresholds of the truck are calculated, and the driving state of the truck in a sharp turn is divided into 3 types according to the speed thresholds. They are as follows:

1. a normal state. When $\mu_y$ satisfies the condition: $\mu_y \leq \mu_{y,critical,rollover}$, where $\mu_{y,critical,rollover}$ represents the maximum lateral acceleration at the moment that wheel lift just occurs, that is, the truck just reaches the risk level 2, combining it and Eq (8), the speed can be expressed as:

$$V \leq \sqrt{127R'(\mu_{y,critical\ rollover} + i_h)} \tag{9}$$

   When the speed meets this condition, the truck is in a normal state. When $\mu_y = \mu_{y,critical,rollover}$, take $V = V_{safe\ state}$, $V_{safe\ state}$ represents the safe speed threshold.

2. a limit state. When $\mu_y$ satisfies the condition: $\mu_{y,critical,rollover} < \mu_y < \mu_{y,rollover}$, combining it and Eq (8), the speed can be expressed as:

$$\sqrt{127R'(\mu_{y,critical\ rollover} + i_h)} < V < \sqrt{127R'(\mu_{y,rollover} + i_h)} \tag{10}$$

   When the speed is in this range, the truck is at risk level 2, but it may still return to the normal driving state. At this time the truck is in a limit state.

3. a dangerous state. When $\mu_y$ satisfies the condition: $\mu_y \geq \mu_{y,rollover}$, combining it and Eq (8), the speed can be expressed as:

$$V \geq \sqrt{127R'(\mu_{y,rollover} + i_h)} \tag{11}$$

   When the speed reaches this condition, the truck may rollover at any time. When the truck just reaches the risk level 3, $\mu_y = \mu_{y,rollover}$, take $V = V_{limit\ state}$, $V_{limit\ state}$ represents the limit speed threshold.

## 3. Results and discussion

### 3.1. Rollover threshold of 4-axle truck based on theoretical model

By Looking up literature, searching the information on Internet and performing related calculations, we get the relevant parameters of the prototype truck. $T$ is equal to 1.874m, $h_g$ is equal to 1.273m when the truck is under a none-loaded condition, $h_g$ is equal to 1.490m when the truck is under an partially-loaded condition, $h_g$ is equal to 1.790m when the truck is under an

**Table 2. Calculation results based on the theoretical model.**

| Formula | Rollover thresholds under the three loading conditions | | |
|---|---|---|---|
| | None-loaded | Partially-loaded | Fully-loaded |
| $0.85\left(\frac{T}{2h_g}\right)$ | 0.616 | 0.526 | 0.438 |

fully-loaded condition. The rollover thresholds of the truck under three different loading conditions are calculated according to the theoretical model, which are shown in Table 2. The unit of rollover threshold in Table 2 is "g", "g" is the acceleration of gravity. The unit of the rollover threshold in the following paper is the same here.

## 3.2. Comparison of simulation test results and theoretical calculation results

**1) Comparison of rollover thresholds based on two methods.** From Table 3, we can find that the rollover threshold obtained by the simulation tests is very close to the rollover threshold calculated by the theoretical model, and the maximum difference is 0.008g. It can be seen that the theoretical calculation results and the simulation tests results are highly consistent, which further shows that the simulation models are reasonable and accurate. The data obtained by simulation method is scientific and reliable.

**2) Dynamic rollover risk levels and rollover margin of each level.** Because the rollover thresholds of trucks based on the two methods are very similar, we will use rollover threshold calculated by the theoretical model to calculate the rollover margin of each risk level. Through the simulation tests, we have got the maximum lateral acceleration of the truck under different dynamic rollover risk levels. According to the definition of rollover margin, the rollover margin is calculated. The results are shown in Table 4. The road features parameters are still as follows: the radius of the circle curve is 5500m, superelevation rate is 0, Longitudinal gradient is 0, friction coefficient of road surface is 0.7. The road characteristic parameters no longer appear in Table 4.

**Table 3. Rollover thresholds based on two methods.**

| Road features | | | | Driver behaviors | | Load parameters | | Rollover threshold by tests | Rollover threshold based on theory | Difference |
|---|---|---|---|---|---|---|---|---|---|---|
| Radius of Circular Curve | superelevation rate | Longitudinal gradient | Friction Coefficient | Speed | Lane change trajectory curve | Load Mass | Height of the center of gravity | | | |
| (m) | (%) | (%) | | (km/h) | Number | (kg) | (m) | (g) | (g) | (g) |
| 5500 | 0 | 0 | 0.7 | 100 | 5 | 0 | 0 | 0.613 | 0.616 | 0.003 |
| | | | | | 4 | 9500 | 1.75 | 0.521 | 0.526 | 0.005 |
| | | | | | 3 | 19000 | 2 | 0.430 | 0.438 | 0.008 |
| 3000 | 2 | 0 | 0.7 | 105 | 5 | 0 | 0 | 0.630 | 0.633 | 0.003 |
| | | | | | 4 | 9500 | 1.75 | 0.539 | 0.543 | 0.004 |
| | | | | | 3 | 19000 | 2 | 0.450 | 0.455 | 0.005 |
| 2000 | 4 | 0 | 0.7 | 105 | 5 | 0 | 0 | 0.645 | 0.650 | 0.005 |
| | | | | | 4 | 9500 | 1.75 | 0.553 | 0.560 | 0.007 |
| | | | | | 3 | 19000 | 2 | 0.465 | 0.472 | 0.007 |
| 1000 | 6 | 0 | 0.7 | 105 | 5 | 0 | 0 | 0.664 | 0.667 | 0.003 |
| | | | | | 4 | 9500 | 1.75 | 0.572 | 0.577 | 0.005 |
| | | | | | 3 | 19000 | 2 | 0.481 | 0.489 | 0.008 |

**Table 4. Dynamic rollover risk levels and rollover margin of each level.**

| Driver behaviors | | Load parameters | | The maximum lateral acceleration | Rollover threshold | Rollover margin | Axle on which the tire has zero vertical reaction | Rollover risk levels |
|---|---|---|---|---|---|---|---|---|
| Speed | Lane change trajectory curve | Load Mass | Height of the center of gravity | | | | | |
| (km/h) | Number | (kg) | (m) | (g) | (g) | (g) | | |
| 105 | 2 | 0 | 0 | 0.488 | 0.616 | 0.128 | Axis 34 | Critical rollover |
| | | 9500 | 1.75 | 0.465 | 0.526 | 0.061 | Axis 34 | Critical rollover |
| | | 19000 | 2 | 0.432 | 0.438 | 0.006 | Axis1~4 | Rollover |
| 100 | | 0 | 0 | 0.460 | 0.616 | 0.156 | Axis 4 | Critical rollover |
| | | 9500 | 1.75 | 0.451 | 0.526 | 0.075 | Axis 34 | Critical rollover |
| | | 19000 | 2 | 0.431 | 0.438 | 0.007 | Axis1~4 | Rollover |
| 90 | | 0 | 0 | 0.397 | 0.616 | 0.219 | None | Safety |
| | | 9500 | 1.75 | 0.402 | 0.526 | 0.124 | Axis 34 | Critical rollover |
| | | 19000 | 2 | 0.378 | 0.438 | 0.060 | Axis 34 | Critical rollover |
| 80 | | 0 | 0 | 0.347 | 0.616 | 0.269 | None | Safety |
| | | 9500 | 1.75 | 0.368 | 0.526 | 0.158 | Axis 4 | Critical rollover |
| | | 19000 | 2 | 0.349 | 0.438 | 0.089 | Axis 34 | Critical rollover |
| 70 | | 0 | 0 | 0.275 | 0.616 | 0.341 | None | Safety |
| | | 9500 | 1.75 | 0.288 | 0.526 | 0.238 | None | Safety |
| | | 19000 | 2 | 0.289 | 0.438 | 0.149 | Axis 4 | Critical rollover |
| 60 | | 0 | 0 | 0.216 | 0.616 | 0.400 | None | Safety |
| | | 9500 | 1.75 | 0.226 | 0.526 | 0.300 | None | Safety |
| | | 19000 | 2 | 0.229 | 0.438 | 0.209 | None | Safety |

Table 4 and Fig 7 show that when the truck travels along the No. 2 lane change trajectory curve, at the speed of 70 km/h, fully-loaded truck just reaches the risk level 2: "Critical rollover", $RM_{ay}$ is equal to 0.149g; at the speed of 80 km/h, partially-loaded just reaches the risk level 2: "Critical rollover", $RM_{ay}$ is equal to 0.158g; at the speed of 100 km/h, none-loaded just reaches the risk level 2: "Critical rollover", $RM_{ay}$ is equal to 0.156g.

We can find that no matter what the loading condition is, when the truck has just reached the risk level 2, that is, the vertical reaction force of one tire is equal to zero, the rollover margins are all approximately equal to 0.15g. According to the definition of rollover margin, there is the following relationship:

$$\mu_{y,critical\ rollover} = \mu_{y,rollover} - 0.15 \tag{12}$$

The rollover thresholds $\mu_{y,rollover}$ under fully-loaded, partially-loaded and none-loaded conditions are known, and the maximum lateral acceleration $\mu_{y,critical\ rollover}$ at the moment when the truck just reaches the risk level 2 can be calculated. Substituting the result into Eq (9), the safe speed threshold $V_{safe\ state}$ can be calculated.

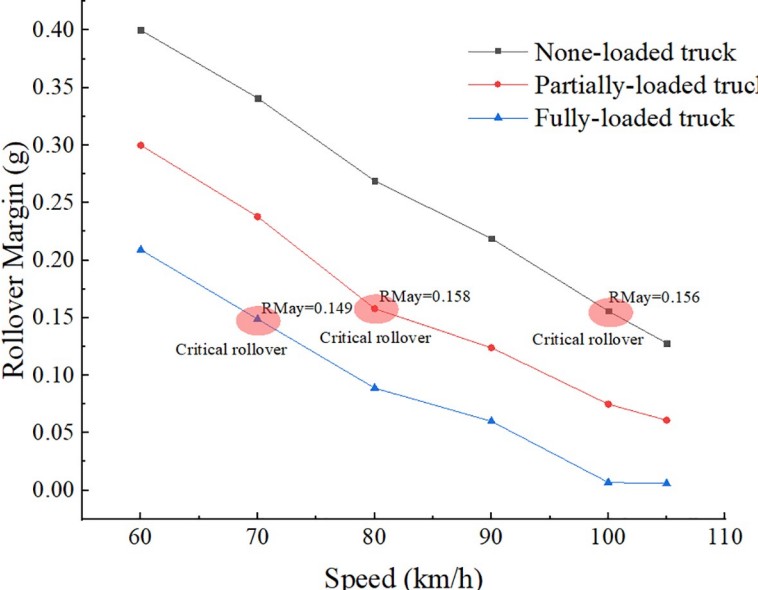

**Fig 7. Rollover margin of the truck.**

## 3.3. Predictions of safe speed threshold and limit speed threshold of the truck in a sharp turn

Because the turning radius of the truck is very small when turning sharply, prediction results when the turning radius less than 250m and the superelevation rate is equal to zero are shown in Fig 8. Because the full-loaded condition is the worst condition, we take the fully-loaded

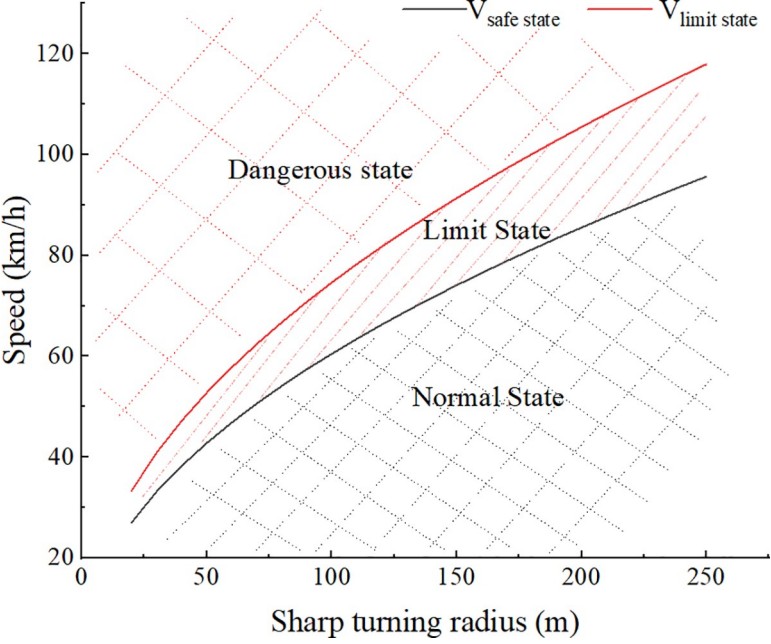

**Fig 8. Predictions of speed thresholds of the truck in a sharp turn.** $V_{safe\ state}$ represents the safe speed threshold; $V_{limit\ state}$ represents the limit speed threshold; Superelevation rate is equal to zero; the loading condition of the truck is fully-loaded.

condition as an example to predict the speed thresholds. The result shows the safe speed threshold and limit speed threshold of fully-loaded truck divide the truck driving state into the normal state, the limit state and the dangerous state. For the safety of driving, the driver should control the speed within the safe speed threshold to keep the truck in a normal state. When the speed exceeds the safe speed threshold but does not exceed the limit speed threshold, the truck is in a limit state, at this time, the driver should be warned. Once the speed exceeds the limit speed threshold, the truck may rollover at any time. This early warning method, compared with the usual safe speed warning method based only on the static rollover threshold, not only allows the driver to understand the truck driving state in more detail and accurately, but also gives the driver sufficient time to adjust the driving state from the limit state back to the normal state, thereby improving the driving safety of the truck.

The prediction results show that as the turning radius increases, both the safe speed threshold and the limit speed threshold gradually increase. When the turning radius is 250m, the limit speed threshold is 118km/h and the safe speed threshold is 97km/h. For the highway with a design speed of 80km/h, the limit minimum value of circular curve radius is 250m, which means that as long as the driving speed selected by the driver is less than or equal to the design speed, the road designed by the existing design theory can ensure the safety of the truck, which is consistent with the study results of Harwood et al. [18–20].

Fig 9 shows that the superelevation has a great influence on the safe speed threshold and the limit speed threshold. The truck turning sharply on a curve section with superelevation. When the truck turns from the outside to the inside of the curve, the superelevation offsets part of the lateral force. When the truck turns from the inside to the outside of the curve, the superelevation does not resist the lateral force, on the contrary, it increases the risk of overturning. At this time, we take the negative of $i_h$ in Eqs (9), (10), and (11). Fig 9 shows that when the turning radius is 200m and the superelevation rate is zero, the safe speed threshold and limit speed threshold are 86km/h and 105 km/h respectively. The truck turns from the outside to the

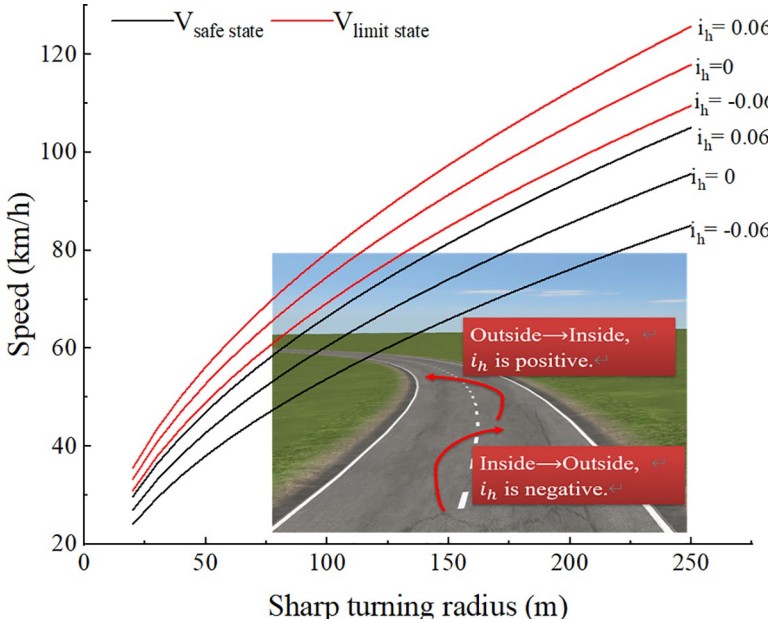

**Fig 9. Predictions of speed thresholds of the truck in a sharp turn with different superelevation rates.** $V_{safe\ state}$ represents the safe speed threshold; $V_{limit\ state}$ represents the limit speed; the loading condition of the truck is fully-loaded.

inside of the curve, and $i_h$ is 0.06. The safe speed threshold and limit speed threshold are 94 km/h and 112 km/h respectively. It can be found that the superelevation can greatly improve the safety of the driving. But when the truck turns from the inner side to the outer side of the curve, $i_h$ is -0.06, and the safe speed threshold and limit speed threshold are 76 km/h and 98 km/h. The superelevation actually reduces the speed thresholds, it is the worst condition to the truck when turning sharply, because of the minimum speed thresholds of the truck under this condition. Therefore, the speed warning is particularly important to the truck under this condition.

## 4. Conclusion

1. with the advantages of advanced simulation software in the field of vehicle dynamics analysis, the comprehensive effects of vehicle design, load and suspension are fully considered, the dynamic process of truck rollover is simulated, the dynamic rollover risk levels of the truck are evaluated, based on dynamic rollover risk levels, the safe speed threshold and limit speed threshold of the truck are predicted. The safe speed obtained by this method is more consistent with the actual situation than that obtained by the statics model, so the accuracy is guaranteed;

2. Speed has a serious impact on the safety of trucks when taking in a sharp turn. The safety speed based on dynamic rollover risk levels proposed in this study is helpful to guide the driver to operate in a sharp turn and ensure the safety of trucks in the process of sharp turn to a greater extent;

3. the safe speed threshold and the limit speed threshold calculated based on the dynamic rollover risk levels in this paper provide important reference for the development of the anti-rollover speed warning system for trucks in the future;

4. this study shows that the simulation software simulates the driver operations with approximating human behavior, and the results are reliable and accurate, the scope of application is wide, this provides new ideas for future traffic safety evaluations.

5. the object in this study is only 4-axle truck, and it can be expanded to study a variety of vehicle types, such as semi-trailer and double trailer in the future.

## Acknowledgments

The authors would like to thank the members of Shaoxing Communications Investment Group Co., Ltd. for their cooperation in the vehicle type survey.

## Author Contributions

**Conceptualization:** Tian Xin, Jinliang Xu.

**Data curation:** Tian Xin, Chao Gao, Zhenhua Sun.

**Formal analysis:** Tian Xin.

**Funding acquisition:** Jinliang Xu.

**Methodology:** Tian Xin.

**Project administration:** Zhenhua Sun.

**Resources:** Jinliang Xu, Zhenhua Sun.

**Software:** Tian Xin.

**Supervision:** Jinliang Xu.

**Writing – original draft:** Tian Xin.

**Writing – review & editing:** Tian Xin, Jinliang Xu.

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
