## [Decision Letter · Decision Letter 0]

29 Jun 2021

PONE-D-21-14975

Research on the Speed Thresholds of Trucks in a Sharp Turn Based on Dynamic Rollover Risk Levels

PLOS ONE

Dear Dr. Xin,

Thank you for submitting your manuscript to PLOS ONE. After careful consideration, we feel that it has merit but does not fully meet PLOS ONE’s publication criteria as it currently stands. Therefore, we invite you to submit a revised version of the manuscript that addresses the points raised during the review process.

We look forward to receiving your revised manuscript.

Kind regards,

Feng Chen

Academic Editor

PLOS ONE

Journal Requirements:

[This research was supported by Scientific Research Project of Zhejiang Provincial

Department of Transportation to JX (Grant No. 2020025).The specific role of these author is articulated in the‘author contributions’ section. There was no additional external funding received for this study.].    

We note that one or more of the authors are employed by a commercial company: Shaoxing Communications Investment Group Co.,Ltd. and China State Construction Silkroad Construction Investment Group Co.,Ltd.

3. We note you have included a table to which you do not refer in the text of your manuscript. Please ensure that you refer to Table 2 in your text; if accepted, production will need this reference to link the reader to the Table.

Reviewers' comments:

Reviewer's Responses to Questions

**Comments to the Author**

1. Is the manuscript technically sound, and do the data support the conclusions?

Reviewer #1: Partly

Reviewer #2: Yes

2. Has the statistical analysis been performed appropriately and rigorously? 

Reviewer #1: No

Reviewer #2: Yes

3. Have the authors made all data underlying the findings in their manuscript fully available?

Reviewer #1: Yes

Reviewer #2: Yes

4. Is the manuscript presented in an intelligible fashion and written in standard English?

Reviewer #1: No

Reviewer #2: Yes

5. Review Comments to the Author

Reviewer #1: Overall, the methodology sounds reasonable although the writing of the paper is quite confusing and in my opinion all of them have to be clarified in order to move forward in the publication process.

• The comparison of simulation test results and theoretical results are performed for radius of curve equal to 5500m, super elevation rate equals to 0%, longitudinal gradient equals to 0%, coefficient of friction 0.70 and speed equals to 100 km. How about the assessment for higher values of super elevation rate i.e., more than 0% in conjunction with speed exceeding 100km/h. Authors should perform more experiments with different values and compare those results with theoretical.

• In line 514, again the authors concluded that the results are reliable and accurate and the scope of application is wide. However, I am still concerned about the limitation of different parameters. Why did the authors consider only flexible pavements (with friction coefficient equals to 0.70) and not rigid pavement?

• Change reference style and make it consistent.

Reviewer #2: The topic of this paper is interesting and important. The methods sound. The results are meaningful and useful. There are some suggestions to improve this paper.

1. The reference style of this paper could be improved. For example, "Darren J. Torbic et al. [33]." could be "Torbic et al. [33]."

2. More references about truck rollover risk are needed. For example, the following one.

[1] Reliability-based assessment of vehicle safety under adverse driving conditions, Transportation Research Part C: Emerging Technology, 18, 507-518.

3. "roll over" could be "rollover".

6. PLOS authors have the option to publish the peer review history of their article (what does this mean?). If published, this will include your full peer review and any attached files.

Reviewer #1: No

Reviewer #2: No

---

## [Author Response · Author response to Decision Letter 0]

7 Jul 2021

Dear Editor Feng Chen and Reviewers,

Thanks very much for taking your time to review this manuscript. I really appreciate all your comments and suggestions! Please find my itemized responses in below and my revisions/corrections in the re-submitted files.

Thanks again!

Response to the Suggestions from editor:

Response: Thank you for the suggestion of file naming. We have made correction according the PLOS ONE style templates. Line4-8, the title of "Research on the Speed Thresholds of Trucks in a Sharp Turn Based on Dynamic Rollover Risk Levels" were corrected as "Research on the speed thresholds of trucks in a sharp turn based on dynamic rollover risk levels".

2. We note that one or more of the authors are employed by a commercial company: Shaoxing Communications Investment Group Co.,Ltd. and China State Construction Silkroad Construction Investment Group Co.,Ltd.

Response: Sorry for the trouble. Through verification, we found that Yafei Liu who is employed by China State Construction Silkroad Construction Investment Group Co., Ltd. played no role in the completion of the paper. After consultation with other authors, all authors agreed to remove Yafei Liu from the author list. Line 11,16-18, the author list was corrected. We are very apologized for the inconvenience caused by the changes to authorship due to negligence. I have updated author roles in the Author Contributions section of the online submission form.

Funding Statement:

This research was supported by Scientific Research Project of Zhejiang Provincial Department of Transportation (No. 2020025) to JX and ZS. This project was completed by the cooperation between Chang'an University and Shaoxing Communications Investment Group Co., Ltd. The funder provided support in the form of salaries for authors, but did not have any additional role in study design, data collection and analysis, decision to publish, or preparation of the manuscript. The specific role of these author is articulated in the ‘author contributions’ section. 

3.Please also provide an updated Competing Interests Statement declaring this commercial affiliation along with any other relevant declarations relating to employment, consultancy, patents, products in development, or marketed products, etc. 

Response: 

Competing Interests Statement:

There are no patents, products in development or marketed products association with this research to declare. This does not alter our adherence to PLOS ONE policies on sharing data and materials.

An updated Funding Statement and Competing Interests Statement are included in the cover letter.

4. We note you have included a table to which you do not refer in the text of your manuscript. Please ensure that you refer to Table 2 in your text; if accepted, production will need this reference to link the reader to the Table.

Response: Line411-413, the statement of "The rollover thresholds of the truck under three different loading conditions are calculated according to the theoretical model, which are shown in Table 2. " was added. Line414,the statement of "the table" was corrected as "table 2".

Response to the reviewer’s comments:

Reviewer #1:

1. The comparison of simulation test results and theoretical results are performed for radius of curve equals to 5500m, superelevation rate equals to 0%, longitudinal gradient equals to 0%, coefficient of friction 0.70 and speed equals to 100 km. How about the assessment for higher values of super elevation rate i.e., more than 0% in conjunction with speed exceeding 100km/h. Authors should perform more experiments with different values and compare those results with theoretical.

Response: According to the international standard, the minimum radius of the circular curve without setting superelevation under a design speed of 120km/h is 5500m, so we set a radius of 5500m and superelevation rate of 0%. In order to solve the reviewer’s doubt about whether simulation test results and theoretical results are consistent under the conditions of higher superelevation rate and higher speed, more experiments were supplemented. In the supplementary experiments, more superelevation and radius conditions based on design specification for highway alignment(JTG D20-2017) were added, which were superelevation rate equals to 2%( radius of curve equals to 3000m)、superelevation rate equals to 4% ( radius of curve equals to 2000m)、superelevation rate equals to 6%( radius of curve equals to 1000m), and longitudinal gradient is still 0%. Since the maximum speed of the selected typical truck is 105km/h, the speed in the supplementary experiments was set to 105km/h. The results of the supplementary experiments were shown in modified Table 3. The results of the supplementary experiments show simulation test results and theoretical results are still very consistent under the conditions of higher superelevation rate and higher speed. Correspondingly, Table 1 was also been modified, which added new scenes, and at this time, Line329-330, the statement of "Scenes 31-39 are used to test the rollover thresholds of the truck at a speed of 105km/h under different superelevation rates. "was added.

2. In line 514, again the authors concluded that the results are reliable and accurate and the scope of application is wide. However, I am still concerned about the limitation of different parameters. Why did the authors consider only flexible pavements (with friction coefficient equals to 0.70) and not rigid pavement?

Response: The supplementary experiments were added for verification. The results of the supplementary experiments show simulation test results and theoretical results are still very consistent under the conditions of higher superelevation rate and higher speed. The establishment of the road model requires the completion of road alignment modeling and pavement modeling with TruckSim, and pavement modeling only needs to input the road friction coefficient, so the software modeling cannot distinguish the performance difference between flexible roads and rigid roads. We study the road conditions in good weather. For new roads, under dry conditions, the road friction coefficient of asphalt concrete pavement and cement concrete pavement is 0.7~0.85. Therefore, it is set to 0.7 in the paper.

3. Change reference style and make it consistent.

Response: Thank you for your comments on the reference style. The style of all references was been carefully checked and revised. 

Reviewer #2:

1. The reference style of this paper could be improved. For example, "Darren J. Torbic et al. [33]." could be "Torbic et al. [33]."

 Response: Thank you for your comments on the reference style. The style of all references was been carefully checked and revised. Line 179, "Darren J. Torbic et al. [33]." was corrected as "Torbic et al. [33]."

2. More references about truck rollover risk are needed. For example, the following one.

 [1] Reliability-based assessment of vehicle safety under adverse driving conditions, Transportation Research Part C: Emerging Technology, 18, 507-518.

Response: Line 545-546, the reference of "Reliability-based assessment of vehicle safety under adverse driving conditions, Transportation Research Part C: Emerging Technology, 18, 507-518. "was added.

3. "roll over" could be "rollover".

Response: Line97,127,129,240,257,268,402,467,the statement of "roll over" was corrected as "rollover".

The above is all the responses to the comments of the editor and reviewers. Thanks again to the editor and reviewers for your valuable suggestions and comments.

We hope that the revised manuscript is accepted for publication in the Journal of PLOS ONE. 

Best regards! 

Sincerely,

Tian Xin

---

## [Decision Letter · Decision Letter 1]

20 Jul 2021

PONE-D-21-14975R1

Research on the speed thresholds of trucks in a sharp turn based on dynamic rollover risk levels

PLOS ONE

Dear Dr. Xin,

Thank you for submitting your manuscript to PLOS ONE. After careful consideration, we feel that it has merit but does not fully meet PLOS ONE’s publication criteria as it currently stands. Therefore, we invite you to submit a revised version of the manuscript that addresses the points raised during the review process.

We look forward to receiving your revised manuscript.

Kind regards,

Feng Chen

Academic Editor

PLOS ONE

Journal Requirements:

Reviewers' comments:

Reviewer's Responses to Questions

**Comments to the Author**

1. If the authors have adequately addressed your comments raised in a previous round of review and you feel that this manuscript is now acceptable for publication, you may indicate that here to bypass the “Comments to the Author” section, enter your conflict of interest statement in the “Confidential to Editor” section, and submit your "Accept" recommendation.

Reviewer #1: All comments have been addressed

2. Is the manuscript technically sound, and do the data support the conclusions?

Reviewer #1: Yes

3. Has the statistical analysis been performed appropriately and rigorously? 

Reviewer #1: Yes

4. Have the authors made all data underlying the findings in their manuscript fully available?

Reviewer #1: No

5. Is the manuscript presented in an intelligible fashion and written in standard English?

Reviewer #1: Yes

6. Review Comments to the Author

Reviewer #1: The authors while responding have quoted " According to International Standards,,...", which they should avoid and explicitly mention the codes/standard. Its better to revise the text accordingly.

7. PLOS authors have the option to publish the peer review history of their article (what does this mean?). If published, this will include your full peer review and any attached files.

Reviewer #1: No

---

## [Author Response · Author response to Decision Letter 1]

2 Aug 2021

Dear Editor Feng Chen and Reviewers,

Thanks very much for taking your time to review this manuscript. I really appreciate all your comments and suggestions! Please find my itemized responses in below and my revisions/corrections in the re-submitted files.

Thanks again!

Response to the Suggestions from editor:

1.Please review your reference list to ensure that it is complete and correct. If you have cited papers that have been retracted, please include the rationale for doing so in the manuscript text, or remove these references and replace them with relevant current references. Any changes to the reference list should be mentioned in the rebuttal letter that accompanies your revised manuscript. If you need to cite a retracted article, indicate the article’s retracted status in the References list and also include a citation and full reference for the retraction notice.

Response: Thank you for the suggestion of reference format. According to the PLOS ONE requirements. The format of each reference has been corrected and completed in line 523-680 of " Revised Manuscript with Track Changes". There is no retracted paper cited in the reference.

2. While revising your submission, please upload your figure files to the Preflight Analysis and Conversion Engine (PACE) digital diagnostic tool, https://pacev2.apexcovantage.com/. PACE helps ensure that figures meet PLOS requirements. 

Response: We have uploaded the figure files to the PACE, and completed the figure files modification.

Response to the reviewer’s comments:

Reviewer #1:

1.The authors while responding have quoted " According to International Standards...", which they should avoid and explicitly mention the codes/standard. It’s better to revise the text accordingly.

Response: Thank you for the suggestion. The expression of "According to International Standards..." is not specific and should be avoid, which is revised as " According to AASHOTO,2018 and MOT, 2017".

where: AASHOTO,2018-American Association of State Highway and Transportation Officials. A Policy on Geometric Design of Highways and Streets. 7th Ed. Washington, D.C. 2018.

MOT, 2017- Ministry of transport of the People’s Republic of China (MOT). Design Specification for Highway Alignment, JTG D20-2017.1st ed. Beijing: China Communications Press; 2017.

The above is all the responses to the comments of the editor and reviewers. Thanks again to the editor and reviewers for your valuable suggestions and comments.

We hope that the revised manuscript is accepted for publication in the Journal of PLOS ONE. 

Best regards! 

Sincerely,

Tian Xin

---

## [Editor Report · Decision Letter 2]

4 Aug 2021

Research on the speed thresholds of trucks in a sharp turn based on dynamic rollover risk levels

PONE-D-21-14975R2

Dear Dr. Xin,

We’re pleased to inform you that your manuscript has been judged scientifically suitable for publication and will be formally accepted for publication once it meets all outstanding technical requirements.

Kind regards,

Feng Chen

Academic Editor

PLOS ONE
---

## [Editor Report · Acceptance letter]

12 Aug 2021

PONE-D-21-14975R2 

Research on the speed thresholds of trucks in a sharp turn based on dynamic rollover risk levels 

Dear Dr. Xin:

I'm pleased to inform you that your manuscript has been deemed suitable for publication in PLOS ONE. Congratulations! Your manuscript is now with our production department. 

Kind regards, 

on behalf of

Dr. Feng Chen 

Academic Editor

PLOS ONE